# A Study on the Robustness of a DNN Under Scenario Shifts for Power Control in Cell-Free Massive MIMO

**DOI:** 10.3390/s25133845

**Published:** 2025-06-20

**Authors:** Guillermo García-Barrios, Manuel Fuentes, David Martín-Sacristán

**Affiliations:** 15G Communications for Future Industry Verticals S.L. (Fivecomm), Camí de Vera s/n (6D Building), 46022 Valencia, Spain; manuel.fuentes@fivecomm.eu; 2iTEAM Research Institute, Universitat Politècnica de València, 46022 Valencia, Spain; damargan@iteam.upv.es

**Keywords:** cell-free massive MIMO, power control, deep neural networks, robustness, spectral efficiency, 6G wireless

## Abstract

The emergence of 6G wireless networks presents new challenges, for which cell-free massive MIMO combined with machine learning (ML) offers a promising solution. A key requirement for practical deployment is the generalizability of ML models—their ability to maintain robust performance across varying propagation conditions, user distributions, and network topologies. However, achieving generalizability typically demands large, diverse training datasets and high model complexity, which can hinder practical feasibility. This study analyzes the robustness of a low-complexity deep neural network (DNN) trained for power control under a single network configuration. The model’s robustness is assessed by testing it across a wide range of unseen scenarios, including changes in the number of access points, user equipment, and propagation environments. The DNN is trained to emulate three power control schemes: max-min spectral efficiency (SE) fairness, sum SE maximization, and fractional power control. To rigorously evaluate robustness, we compare the cumulative distribution functions of performance metrics quantitatively using the Kolmogorov–Smirnov test. Results show strong robustness, particularly for the sum SE scheme, with D statistics below 0.05 and *p*-values above 0.001. This work provides a reproducible framework and dataset to support further research into practical ML-based power control in cell-free massive MIMO systems.

## 1. Introduction

The deployment of fifth generation (5G) networks is currently underway on a worldwide scale. Nevertheless, the landscape of communication technologies continues to evolve, with the emergence of beyond-fifth generation (B5G) and, most importantly, sixth generation (6G) systems. The forthcoming networks are designed to achieve higher data rates while prioritizing low latency and environmentally sustainable wireless communications. To tackle the most recent challenges, a novel technology that combines the benefits of ultra-dense cellular networks with multiple-input multiple-output (MIMO) systems has emerged: cell-free massive MIMO.

In the dynamic field of wireless communication, cell-free networks emerge as a promising solution, surpassing traditional cellular networks in several critical dimensions. Empirical evidence underscores their ability to enhance the signal-to-noise ratio (SNR), leading to reduced SNR variations and improved interference management [1,2,3,4,5]. These networks achieve this by collaboratively serving users through a multitude of access points (APs), thereby effectively mitigating interference.

Despite their advantages, the practical implementation of certain network operations remains a challenge. Tasks such as channel estimation, pilot assignment, and power optimization demand substantial computational resources, potentially limiting real-time deployment [4]. Addressing these computational constraints is crucial for unlocking the full potential of cell-free networks in future wireless communication systems.

Machine learning (ML) emerges as a powerful tool to address the complexities previously mentioned. ML techniques exhibit adaptability across various dimensions, including scheduling, network topology, network traffic, and channel states. However, as the number of user devices increases, the complexity of existing algorithms proposed in the literature also grows. While heuristic schemes offer an alternative approach [2], they introduce two critical problems: the design challenge of an optimal heuristic algorithm for the entire network and the substantial gap between heuristic-based solutions and optimal benchmark algorithms. In this context, ML solutions have attracted considerable attention. Several research works have demonstrated that ML approaches can significantly reduce computational costs while maintaining performance levels comparable to those of the optimal algorithms.

The optimization of power control in the uplink (UL) operation remains a critical challenge in wireless communication systems. Researchers have investigated a number of potential solutions to address this problem, with the objective of enhancing system performance and resource utilization while reducing computational time to ensure its practical implementation.

### 1.1. State of the Art

An initial study by D’Andrea et al. proposed a supervised neural network framework in [6]. The objective was two-fold: (i) to reduce the computational overhead associated with sum-rate and minimum-rate maximization power control schemes and (ii) to analyze the impact of shadowing effects and pilot contamination. The results demonstrated that pilot contamination has a negligible impact on the learning capabilities of the proposed neural network. Bashar et al. introduced another supervised approach based on a deep convolutional neural network (CNN) to solve the sum-rate maximization problem [7]. Notably, their approach considered scenarios with non-active users.

The supervised methodologies require the knowledge of the optimal power coefficients for training the model. However, this requirement imposes a significant computational burden due to the extensive simulation costs involved. To overcome this limitation, an unsupervised deep neural network (DNN) was proposed by Rajapaksha et al. [8]. Their focus was on solving the max-min user fairness problem. Notably, this DNN achieved power control optimization approximately 400 times faster than existing techniques. An updated solution extends the previous DNN framework by incorporating online training and hardware impairment analysis [9]. This resulted in a reduction in computational complexity, rendering the solution more practical for real-world deployment. In a comparable study, Zhang et al. explored the integration of max-sum and max-product rate utilities within a DNN architecture [10]. Their efforts resulted in a notable reduction in training time while maintaining estimator performance, with execution speeds up to 15 times faster achieved. In recent research by the same authors [11], they specifically considered pilot contamination scenarios. In this context, each user equipment (UE) was served by a limited number of APs. The authors focused on the max-min fairness optimization scheme, aiming to strike a balance between system efficiency and fairness. Additionally, results obtained in [12] demonstrated a two-to-three-order-of-magnitude advantage in execution time.

Reinforcement learning (RL) offers a number of advantages over traditional ML systems. These include the capacity to evolve and adapt in response to changing circumstances, to make decisions in the face of uncertainty, to scale efficiently, to optimize long-term performance, and to support autonomous operations [13]. The effectiveness of RL in wireless network applications has been well documented in [14]. In this context, four pivotal studies have been identified. The first proposal was a centralized cell-free strategy, which was designed to solve the multi-objective optimization problem of balancing sum rate and fairness. In order to address the non-convex nature of this problem, a twin delayed deep deterministic policy gradient (TD3) algorithm was employed. The results demonstrated that this approach exhibited superior performance in comparison to sequential convex approximation, particularly in terms of sum rate and minimum user rate [15]. A later study built upon this work included analyses of the computational complexity and convergence properties [16]. Subsequently, Braga et al. investigated the integration of pilot-and-data power control for the optimization of max-min SE fairness and sum SE maximization. They proposed a decentralized actor-critic deep RL method with minimal information exchange and mobility scenarios for the UEs. Their approach yielded results that were on comparable levels to those of centralized solutions and demonstrated superior performance with respect to both estimation and efficiency in terms of pilot contamination [17]. Lastly, Zhang et al. proposed a deep deterministic policy gradient (DDPG)-based power control algorithm to solve the sum-rate maximization problem considering scenarios with both static and mobile users. Remarkably, their research surpasses existing benchmark schemes in terms of user quality of service (QoS) constraints and overall sum-rate performance [18]. Finally, a recent solution has been identified in [19]. Although not a machine learning-based approach, the proposed particle swarm algorithm (PSA)-based methodology is notable for its substantial enhancement in system fairness and its reduced computational complexity. This methodology can effectively complement ML models, providing a robust alternative for addressing complex power control challenges in wireless communication systems.

A summary of all the commented research works is presented in Table 1. The following subsection provides further clarification regarding the generalizability.

### 1.2. Problem Statement

A recent white paper by 5G Americas underscores the critical importance of addressing both scalability and *generalizability* in ML for wireless systems [20]. The concept of generalizability, first introduced in the context of 6G by Chaccour et al. [21], refers to the capacity of an ML solution to perform effectively across a diverse range of deployment scenarios, device configurations, and propagation environments. As highlighted in [20], an ideal ML model would offer robust performance across all conceivable network conditions. However, achieving broad generalizability often entails increased model complexity and computational cost, making the trade-off between generalization and practical deployment efficiency a central challenge in ML-based wireless system design.

In the context of cell-free massive MIMO networks, generalizability takes on added importance due to dynamic and heterogeneous operating conditions. These include variations in user density, topology, active AP count, and propagation environments. While recent work using graph neural networks (GNNs) has shown promising generalization across different network sizes and topologies [22,23], such models typically require complex architectures and decentralized training paradigms. In contrast, our work takes a foundational step toward generalizability by focusing on *robustness*—specifically, analyzing whether a model trained on a fixed configuration of UEs and APs can maintain strong performance when applied to different deployment scenarios. This robustness under scenario shifts allows us to understand how well the learned behavior transfers across unseen environments, without the added complexity of fully generalizable models. By identifying where robustness holds or breaks down, we pave the way for more scalable ML-based solutions that can later incorporate broader generalization strategies with informed design choices.

As discussed in Section 1.1, only a handful of studies have explicitly evaluated generalizability or robustness in cell-free ML systems. Bashar et al. [7] trained models on scenarios with 20 UEs and tested them under reduced user counts, observing performance robustness in downscaled configurations. Similarly, Rajapaksha et al. [9] trained on 10 UEs and evaluated on 5, using a fixed AP configuration. Salaün et al. [24] extended this approach by varying both UE and AP counts during evaluation. However, none of these studies explicitly examined the impact of considering different propagation environments or scenario morphologies, nor did they quantify performance beyond task-specific metrics. In our work, we define robustness as the ability of a model trained in one scenario to maintain a similar output distribution—specifically, the spectral efficiency (SE) per UE—when evaluated under different deployment conditions. To rigorously quantify this distributional similarity, we adopt the Kolmogorov-Smirnov (K-S) test, which compares empirical cumulative distribution functions (CDFs) of SE outcomes between matched and mismatched training-evaluation pairs. Our work shifts focus towards evaluating the robustness of a trained model when applied to unseen scenario variations—including changes in propagation conditions and spatial configurations. To this end, we propose a scenario-diverse dataset and introduce a K-S test-based metric to quantify performance stability under such scenario shifts more rigorously.

### 1.3. Contribution

The studies in [7,9,24] have demonstrated limited forms of generalizability in power control for cell-free massive MIMO. However, they do not directly address generalizability as a core research problem, nor do they evaluate models across a broad set of heterogeneous network scenarios. To fill this gap, we evaluate the robustness of the low-complexity DNN proposed in [25] through extensive simulations that vary the number of UEs, APs, and propagation environments (*urban*, *suburban*, *rural*), following [26] guidelines.

To the best of our knowledge, this is the first study that explicitly focuses on assessing the robustness of ML-based power control models under scenario shifts as an initial step toward broader generalizability in the uplink of cell-free massive MIMO systems. The key contributions are as follows:This paper represents the most extensive robustness study to date for DNN-based power control in cell-free massive MIMO. Unlike prior works, we systematically analyze how well a model trained under a specific configuration performs when applied to unseen variations—altering the number of UEs, APs, or the propagation environment. This includes the challenging case of simultaneous variation in both the number of UEs and APs, and testing across heterogeneous environmental conditions.We reuse and re-evaluate a low-complexity DNN previously proposed in [25], designed to replicate the behavior of three well-known power control strategies: sum SE maximization, max-min SE fairness, and fractional power control (FPC). A novel aspect of this work is the adaptation of the DNN architecture to handle variations in the number of UEs, which affects both the model’s input and output dimensions. This architectural adjustment enables us to evaluate robustness when the number of UEs varies—an aspect not addressed in prior work.To assess robustness beyond standard performance metrics, we compare the empirical CDFs of the DNN and the original algorithms for UE-level SE. This provides a deeper insight into the model’s ability to mimic the baseline distribution under shift conditions. We show that the DNN replicates the sum SE scheme effectively across all variations and the FPC behavior for individual changes in APs or UEs. This distinction underscores the comprehensive adaptability of the proposed DNN model.In addition to a visual comparison of the CDFs, we propose the inclusion of the K-S test in such analyses to gain deeper insights into the correlation between the compared CDFs. We also demonstrate the importance of selecting an appropriate significance level that is well-suited to the data under study.We present a comprehensive set of simulations that led to the creation of a publicly available dataset. This dataset, which covers a wide collection of cell-free massive MIMO network configurations, serves as a valuable resource for the research community. In particular, it enables further exploration and analysis related to power optimization in network systems.

### 1.4. Structure

The paper is structured as follows: Section 2 describes the theoretical framework of the cell-free massive MIMO system under consideration; Section 3 presents the power control schemes considered in the study; Section 4 provides a comprehensive description of the proposed DNN model to solve the power optimization problem; Section 5 describes the dataset used for training and testing the DNN model; Section 6 outlines the methodology used to study the robustness of the proposed model under the different power control schemes; Section 7 presents the results obtained from the experiments; Section 8 summarizes the main findings of the study, and Section 9 suggests potential directions for future research.

### 1.5. Reproducible Research

Results presented in this paper can be reproduced using the dataset and code provided at: https://github.com/Fivecomm/cell-free-power-control-DNN.git (accessed on 18 June 2025).

## 2. System Model

In this study, we investigate a theoretical cell-free massive MIMO scenario. The system comprises *K* single-antenna UEs and *L* APs, each equipped with *N* antennas. The UEs and APs are uniformly and randomly distributed within a square area. A wrapped-around topology is considered to be an effective method for preventing boundary effects [27]. An illustration of the considered cell-free massive MIMO network is provided in Figure 1, which visually summarizes the key components and assumptions of the system model described herein.

### 2.1. Channel Modeling

The assessment of the performance of a cell-free technology necessitates the implementation of a channel model that accurately reflects its behavior. For this analysis, both deterministic and stochastic channel models can be employed. However, the latter are preferred due to their independence from specific scenarios, allowing comprehensive analyses [28].

In the existing literature, the prevailing approach has been to model the channel as a Rayleigh fading channel due to its simplicity [10,24,29,30], assuming a non-line-of-sight (NLoS) environment. Similarly, this study assumes a correlated Rayleigh fading channel model. This choice is motivated by the limitations of uncorrelated Rayleigh fading models, which, although widely used in theoretical investigations, fall short when applied to practical systems. This limitation is supported by numerous measurement campaigns that have consistently shown a statistical correlation between the components of the channel vector [31]. In contrast, correlated Rayleigh fading models offer a more accurate representation of spatial correlation, making them better suited for practical scenarios. This enables the model to consider the elements of the channel vector as complex Gaussian distributed. In this way, the channel between AP *l* and UE *k* is defined as(1)$$h_{kl}∼N_{C}\left(0_{N},R_{kl}\right),$$
where $R_{kl}\in C^{N\times N}$ is the spatial correlation matrix between AP *l* and UE *k*. This matrix estimation is discussed in [28] (Section 3.3.3). On the one hand, the complex Gaussian distribution represents small-scale motions of different objects (transmitters, receivers, etc.) in the propagation environment, whose effect is known as random *small-scale fading*. On the other hand, the correlation matrix represents information regarding geometric spatial channel correlation, antenna gains, path loss, and shadowing [28]. This information is contained in the so-called *large-scale fading coefficients* ($\beta_{kl}$) defined as [2](2)$$\beta_{kl}=\frac{1}{N}\mathrm{tr}\left(R_{kl}\right),$$
where $\mathrm{tr}(R_{kl})$ denotes the trace of matrix $R_{kl}$.

### 2.2. Channel Estimation

The APs and UEs can communicate and cooperate over the same time-frequency resources using the time-division duplex (TDD) protocol or operating in the frequency-division duplex (FDD) mode. In this paper, the TDD protocol is employed, which means that each coherence block is used for both uplink and downlink transmission. This is the optimal approach for estimating the channel, whereby only pilots are transmitted in the uplink, thus avoiding the need for additional channel state information (CSI) exchange [4].

The TDD communication protocol is widely adopted in research studies [8,18] and commercial solutions, where the time-frequency grid is divided into coherence blocks of $\tau_{c}$ transmission symbols, each containing three types of symbols: uplink pilots ($\tau_{p}$), uplink data ($\tau_{u}$), and downlink data ($\tau_{d}$). The data is transmitted at different times, with the pilot signal being sent before the downlink information [2].

In the uplink phase, pilot sequences are transmitted from users to the APs to estimate the channel. To prevent transmission interference, it is preferable that every user utilize a pilot sequence that is orthogonal to the pilots of the rest of UEs. Nevertheless, as the coherence block presents a finite length ($\tau_{c}=\tau_{p}+\tau_{u}+\tau_{d}$), a limited number of mutually orthogonal pilots can be assigned. This fact forces the reuse of pilots between UEs, resulting in the appearance of interference. This is commonly known as *pilot contamination*. In order to avoid interference, a proper pilot assignment algorithm must be selected carefully. There are different types that were already summarized in [4] (Section 3.4.2). In this case, the assignment of the selected pilot signal follows the *master AP* technique, wherein each AP serves a specific subset of UEs [2].

With regard to the selection of the channel estimator, the minimum mean-squared error (MMSE) is the best option when having complete statistical knowledge [5]. This estimator minimizes the mean squared error of the difference between the estimated channel and the real one. Other channel estimation schemes are defined in the literature. One example is the least-square scheme, which was shown to present lower uplink and downlink data rates compared to MMSE [32,33]. Others can be advantageous when handling insufficient statistical knowledge or for reducing computing complexity. This is useful when having a large number of UEs, which is not the case with a cell-free approach [2]. Therefore, MMSE is a sufficient optimal choice.

### 2.3. Uplink Data Transmission

In the uplink transmission, a signal $s_{k}\in C$ is transmitted from user *k* with a power $p_{k}$ and received at an AP *l* with a noise $n_{l}∼N_{C}\left(0,\sigma_{ul}^{2}I_{N}\right)$, where $\sigma_{ul}^{2}$ is the receiver noise power and $I_{N}$ is the identity matrix. The received signal $y_{l}^{ul}\in C^{N}$ expression is(3)$$y_{l}^{ul}=\sum_{k=1}^{K}h_{kl}s_{k}+n_{l}.$$

There are different types of receive combining schemes, where the most commonly used in the literature is the MMSE [7,8,11], which is applicable for a centralized network operation. The research work presented in [6] employed a comparable scheme for a distributed operation, namely the local MMSE (L-MMSE). Neither is a scalable solution, as their complexity increases with the number of UEs [2]. In this paper, the receive combining process utilizes nearly optimal large-scale fading decoding and local partial MMSE (LP-MMSE), a modified version of the L-MMSE proposed as a scalable alternative in [2].

### 2.4. Scalability and Resource Constraints

In a cell-free massive MIMO scenario, the deployment of an AP has a finite capacity in terms of fronthaul and computational processing. Specifically, each AP must be capable of performing signal processing tasks, including channel estimation, data reception and transmission, power optimization, and fronthaul signaling. According to that, a cell-free massive MIMO network can be considered scalable if the computational complexity and resource demands remain finite for each AP when the number of UEs approaches infinity [2].

In consideration of the aforementioned factors, the selected methods for channel estimation, pilot assignment, receive combining, and transmit precoding have been chosen to ensure scalability.

### 2.5. Distributed Operation

This paper considers the distributed cell-free network framework proposed in [2]. Unlike centralized systems, distributed architectures exhibit higher computational complexity due to decentralized processing. However, they offer advantages such as reduced fronthaul signaling and quantization distortion mitigation. To achieve optimal scalability, it is important to emphasize the importance of adding new APs without requiring central processing unit (CPU) upgrades. Each AP is equipped with a local processor, enabling seamless expansion. Furthermore, distributed power allocation strategies, as proposed in [30], facilitate the minimization of large-scale fading (LSF) coefficients exchange between APs.

### 2.6. Number of Antennas per AP

A network comprising a large number of single-antenna APs increases network density, thereby reducing the user-to-antenna distance. This is advantageous for centralized operations as it mitigates interference. However, this is not the case for distributed operations, as the aforementioned configuration only benefits UEs with weak channel conditions. Conversely, having fewer multi-antenna APs enhances UEs with optimal channel conditions. Based on the preceding statement, the selection of a distributed operation, and the analysis conducted in [2], the multi-antenna AP case is adopted in this work.

## 3. Power Control Schemes

The SE is a key parameter for evaluating a communication system. The SE of the channel of user *k* is calculated from the signal-to-interference-plus-noise ratio (SINR) as follows:(4)$${\mathrm{SE}}_{k}=\log_{2}\left(1+{\mathrm{SINR}}_{k}\right).$$

The process of determining the optimal transmission power levels for all users during UL operation is commonly referred to as *power control*. Each user’s transmission power, denoted as $p_{k}$ where $0\;\leq \;p_{k}\;\leq \;p_{max}$ for k=1,…,K, plays a crucial role in achieving efficient communication.

The effective SINR for the uplink SE of each UE *k* is a function of its associated power coefficients. This relationship is expressed as(5)$${\mathrm{SINR}}_{k}\left(p\right)=\frac{b_{k}p_{k}}{c_{k}^{T}p+\sigma_{k}^{2}},$$
where $p={\left[p_{1}\ldots p_{K}\right]}^{T}$ vector represents the UL transmitted powers for *K* users, $b_{k}\geq 0$ characterizes the average channel gain for the desired signal for the *k*-th UE, which reflects the quality of the link between the UE and serving APs. The vector $c_{k}={\left[c_{k1}\ldots c_{kK}\right]}^{T}\in R^{K}\geq 0$ captures the average channel gains associated with interfering signals. Each $c_{kj}$ represents the channel gain for the interfering signals. Lastly, the term $\sigma_{k}^{2}\geq 0$ accounts for the effective noise for the *k*-th user. The aforementioned parameters represent a concise formulation of a more intricate operation, which takes into account the deployment of multiple antennas at the APs, as detailed in [2] (Section 7.1.1).

From the different power control optimization techniques, the most frequently used are sum SE maximization, max-min SE fairness, and FPC [4].

### 3.1. Max-Min SE Fairness

In cell-free networks, achieving fairness in terms of power control is a fundamental goal. The max-min SE fairness objective aims to ensure uniform coverage across the entire network region. Specifically, it seeks to maximize the minimum SE among all UEs. This benefits a reduced number of UEs while compromising the overall system performance. The max-min SE fairness is defined as(6)$$\underset{p\geq 0_{K}}{\mathrm{maximize}}\min_{k\in \{1,\ldots ,K\}}{\mathrm{SINR}}_{k}\;\;\mathrm{subject}\;\mathrm{to}\;p_{k}\leq p_{\max}\;k=1,\ldots ,K.$$

While the problem formulation has been conveniently expressed in terms of the SINR for simplicity, its dependence on the SE becomes evident upon closer examination of the mathematical expression in (Equation 4).

### 3.2. Sum SE Maximization

The optimization of cell-free networks frequently involves the maximization of the total SE across all UEs. This approach represents an optimal strategy, particularly in scenarios where individual UEs introduce interference to only a limited subset of other UEs. The problem of sum SE maximization is formally expressed as follows:(7)$$\underset{p\geq 0_{K}}{\mathrm{maximize}}\sum_{k=1}^{K}{\mathrm{SE}}_{k}\;\;\mathrm{subject}\;\mathrm{to}\;p_{k}\leq p_{\max}\;k=1,\ldots ,K.$$

### 3.3. Fractional Power Control

As previously outlined in Section 2.4, scalability remains a significant concern in cell-free networks. In a recent analysis in [2], it was revealed that existing power optimization approaches fall short in achieving scalability. Among these methods, FPC emerged as a heuristic solution that employed power control mechanisms to mitigate path loss variations within individual cells. It is noteworthy that this approach has been successfully extended to cell-free systems, demonstrating a favorable and scalable heuristic trade-off. Furthermore, the flexibility of FPC allows fine-tuning to optimize various utility functions, enhancing its applicability in practical network scenarios [2].

This research incorporates the evaluation of the FPC scheme, which has garnered significant attention due to its remarkable performance, as demonstrated in [2]. The UL transmitted power for each UE *k* is formulated as(8)$$p_{k}=p_{\max}\frac{{\left(\sum_{l\in M_{k}}\beta_{kl}\right)}^{v}}{\max_{i\in S_{k}}{\left(\sum_{l\in M_{i}}\beta_{il}\right)}^{v}},$$
where $M_{k}\subset \left\{1,\ldots ,L\right\}$ represents the subset of APs that serve UE *k*, $\beta_{kl}$ contains the LSF coefficients associated with the communication links, the exponent *v* governs the behavior of the power control mechanism, and $S_{k}$ denotes the subset of UEs that are partially served by the same APs as UE *k*.

## 4. DNN Model

In this section, the low-complexity DNN that we presented in [25] is considered to address power control optimization in a distributed cell-free massive MIMO network. The model was previously benchmarked against existing supervised [6] and unsupervised [10] learning-based approaches, demonstrating superior performance in terms of spectral efficiency and computational efficiency. Notably, it achieved up to a three-orders-of-magnitude reduction in computation time while maintaining high prediction accuracy across various propagation environments and power control objectives. Given these advantages, the same model is adopted here to explore its robustness in new deployment scenarios.

While most existing literature has focused on unsupervised methods to alleviate the computational burden associated with generating accurate power coefficients [8,10,12], we have opted for a supervised approach. This choice avoids the problem of formulating a robust loss function and enables the implementation of a unified model architecture that accommodates all three power control schemes previously discussed.

The design of the DNN is structured into three distinct subsections: justification for input and output data preprocessing, model architecture, and training configuration.

### 4.1. Data Preprocessing

In the research community, LSF coefficients are frequently utilized as input due to their comprehensive representation of factors such as geometric spatial channel correlation, antenna gains, path loss, and shadowing. In particular, the aggregated version of LSF coefficients, as presented in [10,11], was employed:(9)$$B_{k}=\sum_{l=1}^{L}\beta_{kl}.$$

In order to mitigate the substantial magnitude variations (e.g., spanning from $10^{-5}$ to $10^{4}$) of the LSF coefficients, these are converted to decibels (dB). This conversion is uniformly applied to the output power control coefficients as well. Such transformation facilitates the extraction of relevant information from the dataset. Given that the LSF coefficients are distributed according to a normal distribution, these are standardized, ensuring that the mean is zero and the standard deviation is one. With regard to the output powers, normalization is employed in order to represent them as a fraction of the maximum power $p_{\max}$ that is allowed to be transmitted per UE.

### 4.2. Model Architecture

The architecture of the DNN is outlined in Table 2 and visually depicted in Figure 2. The model employs a feedforward neural network structure with fully connected layers. To enhance generalization and mitigate overfitting, a 50% dropout mechanism is incorporated between layers 1 and 2. Dropout regularization has been demonstrated to be an effective method for preventing the excessive reliance on specific neurons during training. Regarding activation functions, the rectified linear unit (ReLU) is employed for the hidden layers. Additionally, a sigmoid activation function is applied to the output layer to ensure output values lie within the [0,1] range. Lastly, the mean absolute error (MAE) loss function is selected as it is particularly advantageous when dealing with non-Gaussian output distributions containing outliers.

The architecture was intentionally kept shallow and lightweight to minimize computational load, aligning with the goal of enabling real-time deployment in practical cell-free massive MIMO systems. Furthermore, as demonstrated in our previous work [25], this design achieves over a 1000-fold reduction in computational complexity compared to traditional optimization algorithms, reinforcing its suitability for time-sensitive applications.

### 4.3. Training Configuration

In existing literature, the Adam optimizer is identified as a promising approach for addressing non-convex optimization challenges [8,11,12]. The learning rate is set at 0.001, and the model is trained for 100 epochs with a batch size of 128. The dataset is partitioned as follows: 90% of the samples constitute the training split (10% of which are reserved for validation), and the remaining 10% serves as the test set.

### 4.4. Computational Efficiency and Performance

The computational complexity and inference time of the proposed DNN model were thoroughly evaluated in our previous work [25]. In that study, the model was trained using 18,000 samples and tested on 2000 samples. The inference was conducted on a standard personal computer (MSI Katana GF66 12UC ) equipped with a 12th Gen Intel(R) Core(TM) i7-12700H 2.70 GHz processor and 16 GB RAM. The average test time per sample was approximately 0.0001 s (i.e., 0.1 ms), representing a reduction in execution time by more than three orders of magnitude compared to optimization-based solutions (e.g., CVX solvers). Since the DNN architecture and training methodology remain unchanged in the current work, we expect similar inference performance. These results underscore the feasibility of real-time deployment in practical wireless systems.

## 5. Dataset

The assessment of the proposed DNN’s robustness demands a comprehensive exploration across a diverse set of scenarios. This entails considering various combinations of parameters, including the number of UEs and APs. Additionally, network configuration, path loss information, and the standard deviation of shadow fading must be taken into account. In order to achieve this, simulated scenarios from [29] (Table II) are considered. Furthermore, specific features are selected based on Radiocommunication Sector of the International Telecommunication Union (ITU-R) documentation [26].

The present study examines three distinct environments: *urban*, *suburban*, and *rural*. These serve as essential testbeds for evaluating the proposed system. In order to characterize the radio channel in an NLoS propagation environment, the following path loss model, defined by the ITU-R, is adopted:(10)$$\begin{array}{cc} PL(d)=\; & 161.04-7.1\log\left(W\right)+7.5\log\left(h\right)-\left(24.37-3.7{\left(h/h_{AP}\right)}^{2}\right)\log\left(h_{AP}\right)\\ \; & +\left(43.42-3.1\log\left(h_{AP}\right)\right)\left(\log\left(d\right)-3\right)+20\log\left(f_{c}\right)-\left(3.2{\left(\log\left(11.75h_{AT}\right)\right)}^{2}-4.97\right), \end{array}$$
where *d* represents the separation between the antennas of the AP and the UE, *W* characterizes the width of the street or propagation environment, *h* denotes the average height of buildings within the vicinity, $h_{AT}$ and $h_{AP}$ determines the elevation of the UE and AP antennas, respectively. All of these variables are measured in meters. The $f_{c}$ operator specifies the operating frequency in gigahertz (GHz). The path loss of each scenario is intricately tied to specific parameter values, as illustrated in Table 3. The symbol σ represents the standard deviation of log-normal shadow fading, expressed in dB. Additional simulation parameters, not explicitly mentioned here, are detailed in Table 4. To clarify, pilot contamination is considered when K>10, as $\tau_{p}=10$.

In this investigation, a total of 12 distinct scenarios are simulated, each characterized by specific configurations outlined in Table 5. As a reminder, *K* denotes the number of UEs, *L* the number of APs, and *N* the number of antennas per AP. To achieve this, a modified version of MATLAB (R2023a, 9.14.0.2337262, Update 5; MathWorks, Natick, MA, USA) code in [34] is employed, which facilitates the reproduction of all figures and results reported in [2]. In order to ensure a comprehensive exploration of the parameter space, 20,000 setups are generated for each scenario. These setups encapsulate various system configurations, including the number of UEs and APs, antenna placements, and network geometry.

The stored values resulting from the simulations correspond to the LSF and power coefficients necessary for training the DNN models. Furthermore, the average channel gain of the desired signal ($b_{k}$), the average channel gains for the respective interfering signals ($c_{k}$), and the effective noise variance ($\sigma_{k}^{2}$) are stored specifically for the testing split. These values are crucial for calculating the SE using the predicted power coefficients by the DNN in the testing phase, in accordance with (Equation 4) and (Equation 5). Additionally, the SE values of the setups from the test split are also included, which are requisite for comparison with the estimated SE values.

## 6. Methodology

The robustness of the proposed DNN model is evaluated for four distinct case studies: varying the number of UEs, varying the number of APs, varying both the number of UEs and APs, and modifying the type of environment (urban, suburban, and rural) for a fixed configuration of UEs and APs. In the last scenario, as well as in the scenario with a fixed number of UEs, the analysis is straightforward. In this case, the model is trained for a specific scenario and then tested under a different one. However, in the other two cases where the number of UEs varies, the analysis becomes more complex. This complexity arises because both the input and output of the model are dependent on the number of UEs.

To address the aforementioned problem, the implemented solution involves training the model for a maximum number of UEs and testing it with either the same or a reduced number. This is achieved by supplying the DNN with almost zero values ($10^{-20}$ in this case) for any missing inputs and disregarding the output values that are not of interest.

As mentioned before, the approach involves training a model with a fixed number of APs and UEs. The model can then be applied to scenarios with a reduced number of elements (either APs or UEs). However, a model demonstrating high robustness is compatible with a broader range of scenarios. To assess this, we evaluate the trained model on a set of representative configurations commonly studied in the literature, introducing controlled scenario shifts in the number of APs, UEs, and propagation environments (urban, suburban, rural). This enables us to explore how far a model can maintain stable performance when applied to mismatched deployment conditions. The results help identify performance boundaries, beyond which robustness degrades and retraining becomes necessary.

The robustness is assessed using two different methods: conducting a visual comparison between CDFs, or performing a numerical comparison using the K-S test. For both methods, an analytical algorithm (be it max-min SE fairness, sum SE maximization, or FPC described in Section 3) is chosen as the baseline. In particular, to address both optimization problems, max-min SE fairness and the sum SE, a CVX convex solver is employed [35]. In the case of the FPC, the problem is solved using the heuristic approach defined in (Equation 8). Specifically, the exponent value was set to v=−0.5, thereby enabling UEs with poor channel conditions to transmit with higher power.

### 6.1. Visual Comparison of CDFs

In the representation of the CDFs, the baseline is compared with two outcomes: one derived from the trained model tested with data of identical configuration (i.e., the same number of APs, UEs, or scenario types such as urban, suburban, or rural); and the other derived from the trained model with data of a different network configuration, which is tested with data of the scenario under consideration.

### 6.2. Kolmogorov-Smirnov Test

In addition to a visual comparison of both CDFs, a numerical comparison is also incorporated. To the best of our knowledge, this represents a pioneering work in this field that integrates this type of analysis. To facilitate this, a K-S test [36] is conducted.

The K-S test, a non-parametric method, is employed to assess the congruence of one- or two-dimensional probability distributions. In this study, our attention is centered on the two-sample K-S test, which is utilized to determine if two samples originate from an identical distribution. This is typified by the *null hypothesis*, which postulates that the samples are extracted from the same probability distribution. The K-S test yields two principal outputs:**D statistic:** It represents the maximum absolute difference between the CDFs of the two samples. It serves to quantify the distance between the empirical distribution functions of the two samples. A value closer to 0 indicates a higher likelihood that the two samples originate from the same distribution.***p*****-value:** The null hypothesis is rejected if the *p*-value is less than the predetermined significance level. This level is typically set at 0.05, implying a 5% risk of erroneously rejecting the null hypothesis when it is indeed true. If the *p*-value falls below 0.05, the null hypothesis is rejected, leading to the conclusion that a significant disparity exists between the two distributions.

In essence, if the D statistic is minimal and the *p*-value exceeds the selected significance threshold, it can be concluded that the null hypothesis cannot be refuted. Conversely, if the D statistic is substantial or the *p*-value is below the chosen significance level, it can be inferred that the null hypothesis is rejected, leading to the conclusion that the two samples come from different distributions.

The selection of an appropriate significance level is crucial for this test, tailored to the specific requirements of the study. Initial experiments identified an optimal significance level of 0.001. This stringent threshold ensures that there is only a 0.1% chance of incorrectly rejecting the null hypothesis. Consequently, this significantly reduces the probability of false positives, thereby minimizing the risk of erroneously concluding that two samples originate from different distributions when, in fact, they do not.

## 7. Experimental Results

This section presents the results regarding the robustness capabilities under the different case studies using the methodology described above. Specifically, the experiments to assess the robustness with respect to the number of UEs and APs were conducted exclusively in an urban environment. To keep things concise, the terms max-min SE fairness and sum SE maximization will be abbreviated as *MMF* and *sum SE*, respectively, in the following subsections. In the same way, when a DNN is trained in a scenario different from the one under evaluation to assess its robustness, we refer to it as *DNN-sc**X***, where ***X*** denotes the identifier of the scenario used during training.

### 7.1. Robustness to Varying Number of UEs

This section investigates the robustness of the proposed DNN-based approach to changes in the number of UEs between training and evaluation phases.

We first evaluate a mismatch case where the model is trained under scenario 4 (K=9, L=32) and tested under scenario 2 (K=5, L=32). Figure 3 presents the CDF of the SE per UE for three different power control schemes. The results show that the DNN trained under scenario 4 (DNN-sc4) performs similarly to the model trained directly under scenario 2 (DNN) across all power schemes. This suggests a strong robustness capability of the model with respect to a reduction in the number of users, even when the training was performed on a higher-density user configuration.

A similar conclusion holds for a second experiment in a denser deployment with 64 APs. Here, the model is trained under scenario 7 (K=18, L=64) and evaluated under scenario 6 (K=9, L=64), representing a reduction of 9 UEs. As shown in Figure 4, the performance in terms of SE CDF remains largely unchanged for the sum SE and FPC schemes. However, for MMF objective, we observe a noticeable performance drop when the model is trained on 18 UEs (DNN-sc7) and tested on only 9 UEs. This performance drop suggests that the MMF objective is more sensitive to variations in the number of users. This is because MMF focuses on maximizing the performance of the user with the lowest SE, and changes in user density can significantly alter the conditions faced by this worst-case user.

To complement the visual observations, Table 6 presents the results of the K-S test for the various training-evaluation combinations. For most configurations, the *p*-values exceed the significance level of 0.001 and the D statistics are small, supporting the null hypothesis that the SE distributions from mismatched training and testing sets are statistically equivalent. This reinforces the robustness of the proposed model across different user densities. An exception is found in the MMF case mentioned above, where the *p*-value falls below the threshold and the D statistic is higher, confirming the reduced robustness in this specific configuration.

Finally, Table 6 includes an additional case where the model is trained under scenario 4 (K=9) and tested under scenario 3 (K=6), both with 32 APs. Here again, the results confirm statistical similarity, supporting the conclusion that the model maintains robust performance across a range of user densities for most optimization objectives.

In summary, the results demonstrate that the proposed DNN model presents strong robustness under changes in the number of active UEs, particularly for sum SE and FPC objectives. Slight limitations are observed under the MMF scheme, where performance is more sensitive to mismatches in user configuration.

### 7.2. Robustness to Varying Number of APs

This subsection evaluates the effect of varying the number of APs on the robustness of the model. Two scenarios are analyzed.

In the first scenario, the model is trained on scenario 2 (with 32 APs) and tested on scenario 1 (with 24 APs), maintaining a constant number of UEs set to 5. As shown in Figure 5, the DNN trained under matching conditions (DNN) and the one trained with a different number of APs (DNN-sc2) exhibit virtually identical performance for both the sum SE and FPC power control strategies. However, a slight discrepancy emerges for the MMF scheme, where the CDFs diverge, indicating some sensitivity to the number of APs.

In the second scenario, the gap in AP count is larger: the model is trained for 64 APs (scenario 6) and tested on 32 APs (scenario 4), with the number of UEs fixed at 9. The results, illustrated in Figure 6, follow the same pattern—high similarity for sum SE and FPC, while the MMF scheme once again shows a notable performance drop.

To quantify these observations, Table 7 presents the results of the K-S test. The test confirms that for the sum of SE and FPC, the *p*-values exceed the 0.001 threshold and D-values remain low, supporting the null hypothesis of similar distributions. In contrast, the MMF results reject the null hypothesis, confirming statistically significant differences between the DNN trained under evaluation conditions and the one trained under different AP counts. These deviations are visualized in Figure 5b and Figure 6b.

This consistent underperformance of the DNN model for MMF, already observed in the earlier subsection, suggests that MMF is inherently more sensitive to system configuration changes. This is likely due to MMF’s focus on maximizing the SE of the weakest user, making it less tolerant to alterations in network topology. Furthermore, similar challenges in learning the MMF allocation through neural networks have been documented in prior works [8,10].

Given these findings, and in order to focus on scenarios where the DNN robustness is stable, we decided to exclude MMF from the remaining experiments.

### 7.3. Robustness to Joint Variations in UEs and APs

In this section, we evaluate the robustness of the proposed DNN model when both the number of UEs and APs vary between training and testing conditions. A total of five experiments are conducted, each corresponding to a different degree of mismatch between training and testing scenarios.

Figure 7 illustrates the performance for one of the closest configurations: the model is trained under scenario 7 (L=64, K=18) and tested under scenario 5 (L=48, K=12). In this case, the CDFs of the SE per UE for both the sum SE and FPC power allocation schemes are very similar, suggesting that the model presents optimal robustness when the variation in system size is moderate.

To complement the visual analysis, the results of the K-S test for all experiments are shown in Table 8. These statistical tests provide a more precise understanding of how differences in system configuration impact model robustness. For the sum SE objective, the null hypothesis (that both distributions come from the same underlying distribution) is satisfied in all cases except one. This indicates that the model is generally robust to moderate changes in both UEs and APs for this power control scheme. In contrast, the FPC objective exhibits poorer robustness, as seen in consistently lower *p*-values and larger D statistics. Notably, the degradation in performance—measured both visually and statistically—correlates with the degree of mismatch in the number of UEs and APs. That is, as the difference between training and testing conditions increases, the likelihood of the DNN maintaining performance decreases.

These findings confirm that while the proposed model demonstrates strong robustness for the sum SE objective under joint variations in UEs and APs, it is more limited under FPC, especially when the system scale changes significantly.

### 7.4. Robustness Across Different Propagation Environments

In the final set of experiments, we analyze the robustness of the proposed DNN model under varying propagation conditions—namely, urban, suburban, and rural environments—while maintaining fixed network configurations. This analysis provides insights into the model’s robustness under diverse large-scale fading conditions typically encountered in practical deployments.

Two network configurations are considered: one with 32 APs and 9 UEs, and another with 64 APs and 18 UEs. For each configuration, the model is trained on data generated from one specific propagation scenario and evaluated across all three. This results in a comprehensive cross-scenario evaluation that encompasses all combinations of training and testing environments.

The results of the K-S test for all experiments are presented in Table 9. The table reveals a clear distinction between the behavior of the model under the two power control schemes considered:**Sum SE:** In all cases, the null hypothesis is satisfied, indicating that the empirical distributions of SE under different propagation scenarios are statistically equivalent. This demonstrates a high degree of robustness and suggests that the DNN trained with the sum SE objective generalizes well across environments with different propagation characteristics.**FPC:** In contrast, the majority of the experiments with FPC do not satisfy the null hypothesis, indicating sensitivity to changes in the propagation environment. However, some specific trends are observed: the most favorable outcomes (i.e., higher *p*-values and lower D-statistics) occur when training and testing involve either the suburban or rural scenario. On the other hand, the least favorable results consistently involve the urban environment, either in training or testing.

These findings suggest that the robustness of the DNN is highly dependent on the power optimization scheme employed. The sum SE scheme enables the model to capture general patterns in the input data that remain valid across environments, resulting in greater generalizability. In contrast, the FPC scheme’s performance degrades with environment mismatch, likely due to its tighter dependency on large-scale fading features, which vary more significantly between propagation scenarios.

Overall, these results confirm that the proposed DNN demonstrates strong robustness across diverse propagation scenarios when trained with the sum SE power scheme. For FPC, generalization is more limited but still acceptable between similar environments such as suburban and rural.

## 8. Conclusions

In this paper, we presented a supervised, low-complexity DNN model for uplink power control in distributed cell-free massive MIMO networks. The proposed model significantly reduces the computational complexity compared to conventional optimization methods while maintaining competitive performance across diverse conditions. Unlike prior efforts that pursue universal generalizability—often requiring complex architectures and large-scale data—our work takes a more pragmatic first step: systematically evaluating the robustness of a fixed-size model under a broad spectrum of realistic deployment variations.

Our analysis considered key sources of scenario variability, including changes in the number of UEs, APs, and propagation environments (urban, suburban, and rural). We evaluated the model under three different power control strategies: sum SE maximization, max-min fairness, and FPC. Results indicate that the model exhibits strong robustness under scenario shifts for the sum SE scheme. In the case of FPC, robustness was observed under isolated changes in UE or AP counts, but performance degraded under joint variation. For environmental shifts, the model maintained stable behavior between rural and suburban settings while exhibiting reduced robustness when the urban scenario was involved.

These findings underscore the practical value of robustness as an intermediate goal. By training a simple model under fixed-size conditions and validating its stable performance across diverse scenarios, we provide a scalable and deployable ML-based solution tailored for real-world wireless networks. This robustness approach enables a more feasible path towards adaptive, generalizable ML models by reducing retraining requirements and architectural complexity in the short term.

Furthermore, our findings indicate that the proposed model maintains robustness primarily under moderate scenario shifts. As the disparity between training and testing conditions grows—particularly when jointly varying AP and UE counts—performance degradation emerges. Rather than pursuing a single, universally generalizable model with increased architectural complexity, we advocate for a more practical alternative: training multiple models tailored to specific operating ranges. At deployment, the model whose training conditions most closely match the current network state can be selected. This strategy preserves low complexity while supporting broad applicability across real-world scenarios.

Finally, we emphasize that relying solely on visual CDF comparisons can be misleading. Incorporating statistical tools such as the K-S test provides a more rigorous and interpretable framework. Additionally, adjusting the significance level for *p*-values proves to be a simple yet effective method to facilitate faster and more reliable performance assessment across diverse scenarios.

From a practical standpoint, the proposed model is intended to be integrated into the CPU of a cell-free massive MIMO system. Since the DNN operates with low complexity and requires only large-scale fading coefficients as input, it can be executed in real time to support fast power allocation decisions. Moreover, due to its robustness across moderate scenario variations, the model does not require frequent retraining, making it well-suited for deployment in networks with dynamic but structured variability—such as urban hotspots, public venues, or rural areas with predictable traffic patterns. This aligns with ongoing efforts in 6G architecture to balance central intelligence with lightweight on-device inference.

## 9. Future Work

While several significant findings have emerged from this study, the topic of generalizability in ML-based power control for cell-free massive MIMO systems remains a fertile ground for further research.

First, a more detailed analysis is warranted to determine the robustness range of the model with respect to variations in the number of APs and UEs. Specifically, it would be valuable to identify the limits within which a model trained on a specific configuration can maintain reliable performance as these network parameters change. Second, although the proposed DNN model exhibits strong robustness for the sum SE maximization scheme, the FPC scheme shows more constrained robustness—particularly when both UEs and APs vary simultaneously—and the max-min fairness objective remains more challenging to replicate accurately. Future work could explore the design or adaptation of alternative neural architectures better suited to capturing the complexities of these scenarios and improving robustness under broader deployment shifts.

Third, the use of the Kolmogorov–Smirnov test showed promising potential in complementing CDF analysis. However, identifying a formal methodology to determine the optimal *p*-value threshold that balances sensitivity and interpretability remains an open challenge. Establishing such a framework would enable more rigorous statistical assessments of model performance.

Fourth, while this study focused on evaluating robustness under scenario shifts—such as changes in topology and propagation conditions—future work could aim to extend these insights toward achieving broader forms of generalizability. One promising direction is task generalization, where a single model is trained to adapt across different optimization goals (e.g., transitioning from sum SE to energy efficiency or latency-aware objectives). Another is architecture generalization, which involves designing size-agnostic models capable of operating across varying numbers of UEs and APs without the need for retraining. Pursuing these directions would help bridge the gap between robust scenario-specific models and fully generalizable, scalable ML-based solutions for wireless systems.

Lastly, this study employed a Rayleigh fading channel in simulations. Future investigations could explore Rician fading environments, which include a dominant line-of-sight (LoS) component and are more representative of realistic urban deployments. Incorporating this into the simulation framework could provide further insight into the model’s ability to generalize under diverse physical-layer conditions, especially in scenarios involving mobility and phase noise.

## Figures and Tables

**Figure 1 sensors-25-03845-f001:**
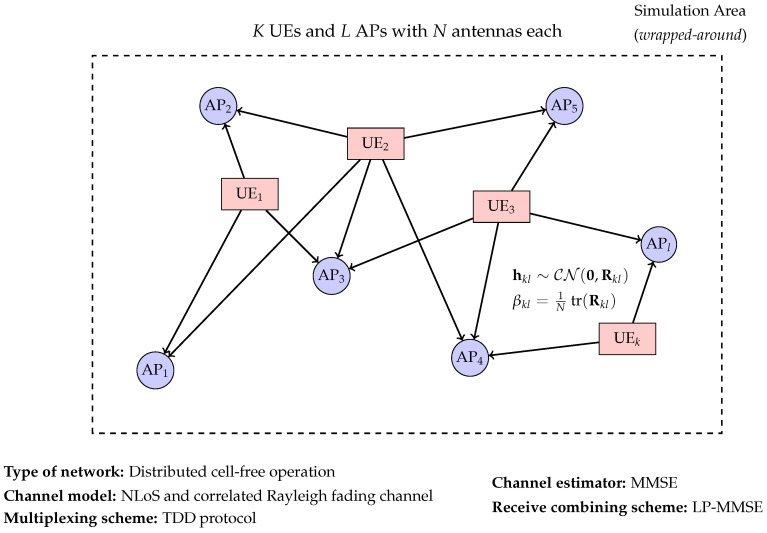
System model for a distributed cell-free massive MIMO network.

**Figure 2 sensors-25-03845-f002:**
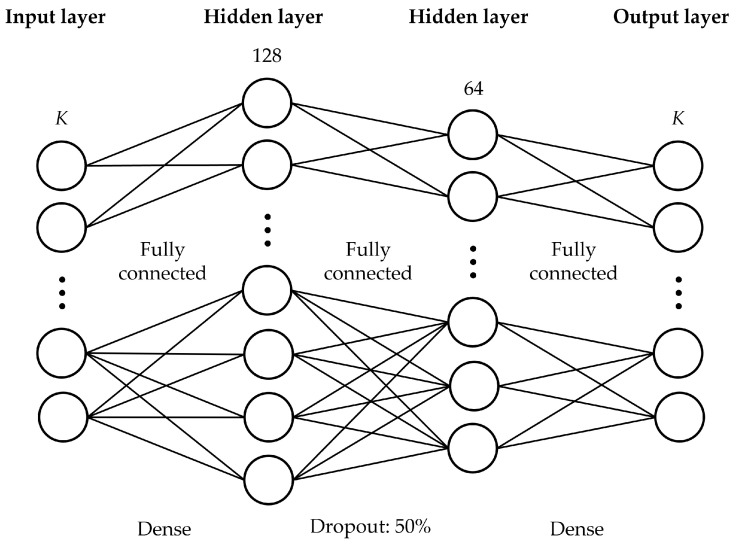
Architecture of the proposed DNN.

**Figure 3 sensors-25-03845-f003:**
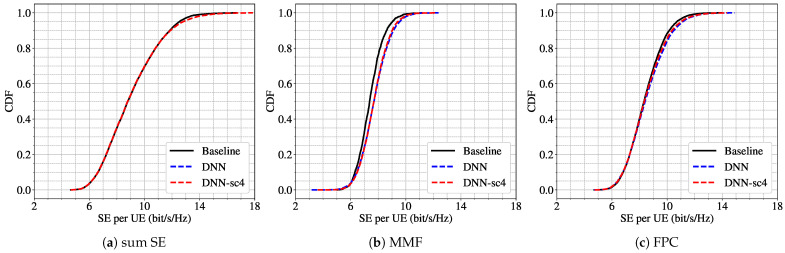
CDF of per-UE SE for scenario 2 (L=32, K=5) using three power control schemes: (**a**) sum SE; (**b**) MMF; (**c**) FPC. Results from the baseline algorithm (black line), the DNN trained for scenario 2 (blue dashed line), and the DNN trained for scenario 4 (L=32, K=9) (red dashed line).

**Figure 4 sensors-25-03845-f004:**
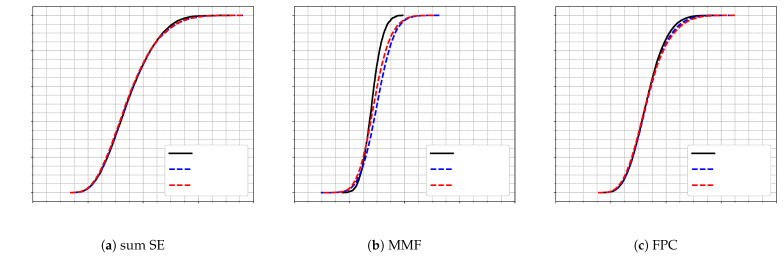
CDF of per-UE SE for scenario 6 (L=64, K=9) using three power control schemes: (**a**) sum SE; (**b**) MMF; (**c**) FPC. Results from the baseline algorithm (black line), the DNN trained for scenario 6 (blue dashed line), and the DNN trained for scenario 7 (L=64, K=18) (red dashed line).

**Figure 5 sensors-25-03845-f005:**
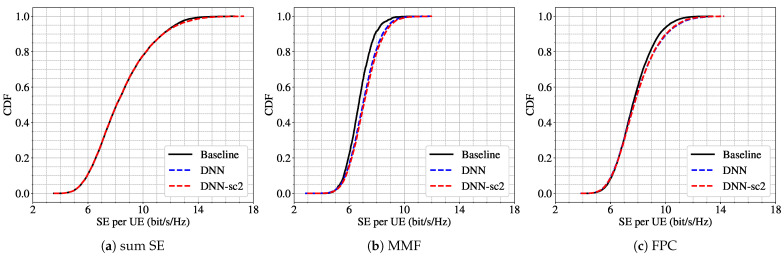
CDF of per-UE SE for scenario 1 (L=24, K=5) using three power control schemes: (**a**) sum SE; (**b**) MMF; (**c**) FPC. Results from the baseline algorithm (black line), the DNN trained for scenario 1 (blue dashed line), and the DNN trained for scenario 2 (L=32, K=5) (red dashed line).

**Figure 6 sensors-25-03845-f006:**
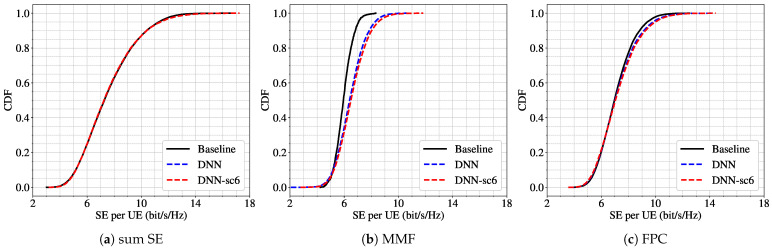
CDF of per-UE SE for scenario 4 (L=32, K=9) using three power control schemes: (**a**) sum SE; (**b**) MMF; (**c**) FPC. Results from the baseline algorithm (black line), the DNN trained for scenario 4 (blue dashed line), and the DNN trained for scenario 6 (L=64, K=9) (red dashed line).

**Figure 7 sensors-25-03845-f007:**
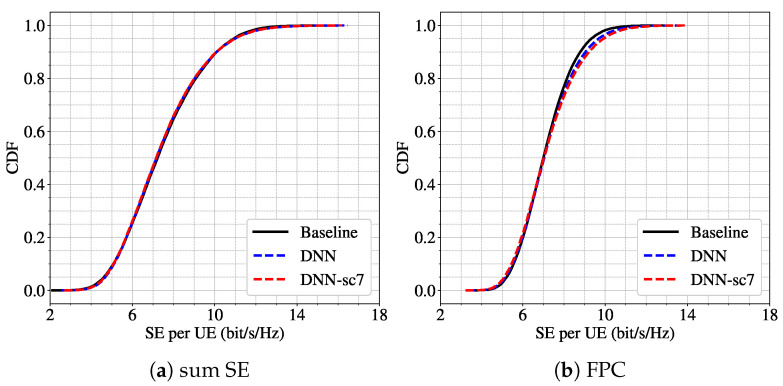
CDF of per-UE SE for scenario 5 (L=48, K=12) using two power control schemes: (**a**) sum SE; (**b**) FPC. Results from the baseline algorithm (black line), the DNN trained for scenario 5 (blue dashed line), and the DNN trained for scenario 7 (L=64, K=18) (red dashed line).

**Table 1 sensors-25-03845-t001:** Summarized research works focused on ML power control.

Paper	Power Scheme	ML Type	Model	Generalization
[6]	Max-sum Max-min	Supervised	DNN	No
[7]	Max-sum	Supervised	CNN	Yes
[8]	Max-min	Unsupervised	DNN	No
[9]	Max-min	Unsupervised	DNN	Yes
[10]	Max-sum Max-product	Unsupervised	DNN	No
[11]	Max-min	Unsupervised	DNN	No
[12]	Max-min	Unsupervised	DNN	No
[15] [16]	Sum-rate fairness trade off	Reinforcement learning	TD3	No
[17]	Max-sum Max-min	Reinforcement learning	DDPG	No
[18]	Max-sum	Reinforcement learning	DDPG	No

**Table 2 sensors-25-03845-t002:** Layout of the proposed DNN.

	Size	Parameters	Activation Function
Input	*K*	-	-
Layer 1 (Dense)	128	(*K* + 1) 128	ReLU
Layer 2 (Dropout: 50%)	64	8256	ReLU
Layer 3 (Dense)	*K*	65K	Sigmoid

**Table 3 sensors-25-03845-t003:** Parameters of path loss models in [26].

Parameter	Urban	Suburban	Rural
*W* (m)	20	20	20
*h* (m)	20	10	5
$h_{AP}$ (m)	20	20	40
$h_{AT}$ (m)	1.5	1.5	1.5
$f_{c}$ (GHz)	2	2	0.45
σ (dB)	6	8	8

**Table 4 sensors-25-03845-t004:** Fixed parameters of simulations.

Parameter	Values
Number of setups	20,000
Number of channel realization	100
Length of the coherence block ($\tau_{c}$)	200 channel uses
Number of pilots per coherence block ($\tau_{p}$)	10
Total UL transmit power per UE	100 mW
Total DL transmit power per AP	200 mW
Decorrelation distance of the shadow fading	9 m
Antenna spacing (in number of wavelengths)	1/2
Noise figure	7 dB
Bandwidth	20 MHz
Azimuth angular standard deviation	15°
Elevation angular standard deviation	15°

**Table 5 sensors-25-03845-t005:** Simulated scenarios.

Scenario	Type	Square Length (m)	L	K	N
1	Urban	500	24	5	4
2	Urban	500	32	5	4
3	Urban	500	32	6	4
4	Urban	500	32	9	4
5	Urban	500	48	12	4
6	Urban	500	64	9	4
7	Urban	500	64	18	4
8	Urban	500	96	30	4
9	Suburban	1000	32	9	4
10	Suburban	1000	64	18	4
11	Rural	5000	32	9	4
12	Rural	5000	64	18	4

**Table 6 sensors-25-03845-t006:** K-S test for evaluating robustness to varying number of UEs.

L	Train	Test	Scheme	*p*-Value	D
32	Sce. 4 (K=9)	Sce. 2 (K=5)	sum SE	1	0.0001
			MMF	0.1	0.02
			FPC	0.03	0.02
		Sce. 3 (K=6)	sum SE	1	$8\times 10^{-5}$
			MMF	0.1	0.02
			FPC	0.07	0.02
64	Sce. 7 (K=18)	Sce. 6 (K=9)	sum SE	0.4	0.01
			MMF	$8\times 10^{-80}$ *	0.1
			FPC	0.001	0.02

The values marked with * reject the null hypothesis (*p*-value < 0.001).

**Table 7 sensors-25-03845-t007:** K-S test for evaluating robustness to varying numbers of APs.

K	Train	Test	Scheme	*p*-Value	D
5	Sce. 2 (L=32)	Sce. 1 (L=24)	sum SE	1	0.0002
			MMF	$2\times 10^{-4}$ *	0.03
			FPC	1	0.01
9	Sce. 6 (L=64)	Sce. 4 (L=32)	sum SE	1	$6\times 10^{-5}$
			MMF	$3\times 10^{-13}$ *	0.04
			FPC	0.008	0.02

The values marked with * reject the null hypothesis (*p*-value < 0.001).

**Table 8 sensors-25-03845-t008:** K-S test for evaluating robustness to varying number of UEs and APs.

Train	Test	Scheme	*p*-Value	D
Sce. 7 (L=64, K=18)	Sce. 4 (L=32, K=9)	sum SE	0.4	0.01
		FPC	$4\times 10^{-5}$ *	0.02
	Sce. 5 (L=48, K=12)	sum SE	0.8	0.01
		FPC	$6\times 10^{-4}$ *	0.02
Sce. 8 (L=96, K=30)	Sce. 4 (L=32, K=9)	sum SE	$7\times 10^{-4}$ *	0.02
		FPC	$2\times 10^{-14}$ *	0.04
	Sce. 5 (L=48, K=12)	sum SE	0.02	0.01
		FPC	$1\times 10^{-10}$ *	0.03
	Sce. 7 (L=64, K=18)	sum SE	0.6	0.01
		FPC	$8\times 10^{-5}$ *	0.02

The values marked with * reject the null hypothesis (*p*-value < 0.001).

**Table 9 sensors-25-03845-t009:** K-S test for evaluating robustness across different propagation environments.

L	K	Train	Test	sum SE		FPC	
				* **p** * **-Value**	**D**	* **p** * **-Value**	**D**
32	9	Sce. 4 (Urban)	Sce. 9 (Suburban)	1	0.0001	$1\times 10^{-11}$ *	0.04
			Sce. 11 (Rural)	1	0.0001	$1\times 10^{-7}$ *	0.03
		Sce. 9 (Suburban)	Sce. 4 (Urban)	1	0.0001	$3\times 10^{-19}$ *	0.05
			Sce. 11 (Rural)	1	0.0001	0.2	0.01
		Sce. 11 (Rural)	Sce. 4 (Urban)	1	0.0001	$5\times 10^{-14}$ *	0.04
			Sce. 9 (Suburban)	1	0.0001	0.09	0.01
64	18	Sce. 7 (Urban)	Sce. 10 (Suburban)	0.1	0.01	$6\times 10^{-12}$ *	0.03
			Sce. 12 (Rural)	0.04	0.01	$9\times 10^{-24}$ *	0.04
		Sce. 10 (Suburban)	Sce. 7 (Urban)	0.2	0.01	$9\times 10^{-19}$ *	0.03
			Sce. 12 (Rural)	1	0.002	$6\times 10^{-5}$ *	0.02
		Sce. 12 (Rural)	Sce. 7 (Urban)	0.09	0.01	$4\times 10^{-49}$ *	0.06
			Sce. 10 (Suburban)	1	0.002	$4\times 10^{-6}$ *	0.02

The values marked with * reject the null hypothesis (*p*-value < 0.001).

## Data Availability

New data were created in this study. Simulation data is available at https://zenodo.org/records/10691343 (accessed on 18 June 2025).

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
