# Peer review of "A Study on the Robustness of a DNN Under Scenario Shifts for Power Control in Cell-Free Massive MIMO"

_sensors, 2025, doi:10.3390/s25133845_

Round 1

Reviewer 1 Report

Comments and Suggestions for Authors

This paper focuses on analyzing the robustness of a low-complexity deep neural network (DNN) trained for power control under a single network configuration. The key idea is interesting. However, following issues need to be address to improve the quality of the paper.

1). In the abstract it is said, " Results show strong robustness, particularly for the sum SE scheme.". It is better to provide some average numerical gains of the proposed approaches.

2). There is no system model diagram in the paper. It would be  better to show the system model considered in a diagram, and include most critical system parameters.

3). This paper considers a low complexity DNN presented in the reference [25]. Why did you specifically consider this model, not much motivations are provided. Compare the results with different approaches.

4). It would be better to draw the neural network architecture as well.

5). How does the power constraints are considered in the DNN model?

6). Most of the results figures cannot be differentiated. Better to zoom of use a specific range.

7). Can you plot some average results, apart from the CDF presented in the results section. Like the SE, for different schemes. 

8). Results should be better explained.

9). The formatting of the paper is poor. Use a consistent formatting, use abbreviations for journals, for conferences use "In Proc. xx".

Reviewer 2 Report

Comments and Suggestions for Authors

The authors have done a great job — the idea is relevant and has great value. It is especially pleasing that both the code and the dataset are posted on GitHub, and everything really runs without problems.

However, there are several wishes:

I would like a more accurate interpretation of the results, especially in cases where the model loses accuracy (for example, with max-min fairness or simultaneous shifts of AP and UE).
Reproducibility is still rather formal — there is not enough information about the inference time, resources, and performance.
The formalization of the "robustness" problem remains quite general. It would be useful to more clearly define the metric and justify the choice of approaches.
I would also like to see a comment on practical application: where and how the model is planned to be used in real systems.
Overall, the work is strong and deserves publication after these points are finalized.

Round 2

Reviewer 1 Report

Comments and Suggestions for Authors

The authors have well addressed all the comments. No further comments from the reviewer.